

# Advancing Arctic sea ice remote sensing with AI and deep learning: now and future

Wenwen Li[1], Chia-Yu Hsu[1], and Marco Tedesco[2, 3]

[1]School of Geographical Sciences and Urban Planning, Arizona State University, Tempe, AZ 85226
[2]Lamont-Doherty Earth Observatory, Columbia University, Palisades, New York 10964
[3]NASA Goddard Institute of Space Studies, New York, New York 10025

**Correspondence:** Wenwen Li (wenwen@asu.edu)

**Abstract.** The revolutionary advances of Artificial Intelligence (AI) in the past decade have brought transformative innovation across science and engineering disciplines. Also in the field of Arctic science, we have witnessed an increasing trend in the adoption of AI, especially deep learning, to support the analysis of Arctic big data and facilitate new discoveries. In this paper, we provide a comprehensive review of the applications of deep learning in sea ice remote sensing domains, focusing on problems such as sea ice lead detection, thickness estimation, concentration, sea ice extent forecasting and motion detection as well as sea ice type classification. In addition to discussing these applications, we also summarize technological advances that provide customized deep learning solutions, including new loss functions and learning strategies to better understand sea ice dynamics. To promote the growth of this exciting interdisciplinary field, we further explore several research areas where the Arctic sea ice community can benefit from cutting-edge AI technology. These areas include improving multi-modal deep learning capabilities, enhancing model accuracy in measuring prediction uncertainty, better leveraging AI foundation models, and deepening the integration with physics-based models. We hope that this paper can serve as a cornerstone in the progress of Arctic sea ice research using AI and inspire further advances in this field.

## 1 Introduction

Arctic sea ice is integral to global ecosystems of the Northern hemisphere as it serves as a vital climate regulator, helping to maintain the Earth's surface temperature by reflecting sunlight back into space. Sea ice also provides essential habitat for numerous wildlife species and plays a key role in global ocean circulation, influencing ocean currents and weather patterns on a global scale. The extent of Arctic sea ice has declined during the past few decades, with an unprecedented rate of nearly 13% per decade estimated at the end of Arctic summer seasons (Lindsey and Scott, 2020). Arctic sea ice has also thinned considerably and shifted towards being younger season after season. According to the IPCC, since 1979, the area covered by 5-year-old ice has decreased by approximately 90% (Pörtner et al., 2019). The loss of Arctic sea ice has resulted in concerning consequences, including rising sea levels and an increase in coastal disasters (Barnhart et al., 2014), altered weather patterns (Labe et al., 2020), disruptions in ecosystems (Post et al., 2013), and impacts on indigenous communities (Hauser et al., 2021). Therefore, it is crucial to develop new monitoring and analytical capabilities to advance our understanding of sea ice dynamics, the changes resulting from recent warming, and their impact on the global environment and climate.



Remote sensing has been used as an important vehicle for collecting spatiotemporal information about sea ice properties and conditions in the polar regions (e.g., Sandven et al., 2023). Traditional methods for remote sensing image analysis in Arctic sea ice research have relied on statistical approaches (Barber and Ledrew, 1991). While popular, these methods also suffer from notable limitations. For example, statistical approaches are sensitive to data quality. Errors and noise in satellite imagery and data value inconsistency among different data sources may lead to inaccuracy in the analytical results (Congalton, 1991).

Meanwhile, statistical approaches are often point-based analysis, and they cannot handle big spatial data which have high spatial and temporal resolutions effectively. In addition, while statistical approaches (for example, ordinary least square regression and Bayesian statistics) are good at understanding the cause-and-effect relationships, their prediction capability for unseen datasets is limited. These approaches often assume certain relationships between the independent and dependent variables, such as linear or logistic relations, they therefore cannot capture complex relationships within the data. In comparison, as a

new kind of statistical method, shallow machine learning, such as random forests and support vector machines, have overcome some of these issues to better handle big and complex datasets, and achieve good performance and efficiency (Waske et al., 2009). However, these methods often require feature engineering, the process to manually extract important features that will affect the outcome from the raw data. The manual process is time-consuming and the incomplete set of selected features will result in a reduced accuracy in the analytical results.

Object-based image analysis (OBIA) is also an important technique in sea ice remote sensing research (Sha et al., 2023). It operates by automatically segmenting images into image regions and then conducting further classification based on predefined rules. This method also involves manual work, which suffers from limited generalizability when applied to diverse study areas and datasets. Fortunately, the advances in artificial intelligence (AI), especially deep learning, have brought fundamental changes by enabling effective data-driven analysis (Li and Hsu, 2022). In particular, the revolutionized development of con-

volutional neural network (CNN) models allows for the automated extraction of both low-level and high-level image features critical for detecting new phenomena and events, as well as classifying different types of objects within an image (Li, 2020). The local, convolutional operations of CNN models also allow for parallel processing, making it possible to handle extensive geospatial data to extract generalizable patterns and trends. In recent years, Arctic researchers have increasingly embraced deep learning technology in sea ice research. Figure 1 provides an overview of the integration and customization of deep learning

models to support Arctic sea ice research.

Excitingly, pioneering research has been found in multiple sea ice applications, such as sea ice lead detection, sea ice thickness estimation, sea ice concentration estimation and forecasting, sea ice motion detection, and sea ice type classification. In addition to CNN models, several deep learning feature extraction backbone models have been developed (Figure 1, Module named "Deep learning (backbone models)"). This includes the multi-layer perceptron (MLP; Bogdanov et al., 2006)

model, which predicts the classes (e.g., sea ice type) of an image pixel or an image patch based on extracted feature. The recurrent neural network (RNN; Medsker and Jain, 2001) is designed for processing time-sequence input for forecasting tasks (e.g., predicting sea ice motions). A transformer-based model uses a self-attention mechanism to capture spatial and temporal data dependencies over a long range, achieving even more powerful pattern analysis and extraction than many CNN models (Vaswani et al., 2017) Transformer-based models were originally designed for processing time-sequence data and have recently





**Figure 1.** Deep learning-informed sea ice research

been adopted in the image analysis and computer vision domain through the development of vision transformers (ViT) and their variants. This new architecture, while computationally intensive, can be adapted for both image segmentation and forecasting tasks. Image segmentation, such as the estimation of sea ice concentration, can also be achieved through an interesting deep learning model architecture called a generative adversarial network (GAN; Goodfellow et al., 2014). Instead of aiming for information extraction and pattern recognition, the GAN is a type of generative AI. It introduces a generative network to generate

image results as close to the expected results as possible and uses a discriminative network to evaluate them. In addition to the visual models, graph convolutional neural networks (GCN; Zhang et al., 2019) have also been employed to use graph theory to guide the topological analysis of key sea ice features, such as sea ice leads, to enhance their detection and movement.

Based on these backbone models, various deep learning architectures (e.g., U-Net) have been applied to support sea ice research based on the problem formulation (Figure 1, Module named "Deep learning applications"). For example, certain

research (e.g., sea ice concentration estimation) can be formulated as a spatial problem aiming to segment remotely sensed





images at a pixel or patch level. Other research can be formulated as a spatial and temporal problem for forecasting based on time-sequence image data. The same research can also be formulated differently and solved using various architectures.

This paper will organize the review according to the application of these deep learning models in different sea ice research areas (Section 2). Differing from other reviews, our paper focuses on AI, particularly the applications of deep learning in various
subareas of sea ice research, offering a detailed analysis of the integration of these two fields. Section 3 further discusses how new learning strategies are customized in the sea ice-deep learning solution framework (Figure 1, Module on "New techniques integrated"). Section 4 presents our perspective on the future advances that can be achieved through a deep integration of AI and sea ice research. Section 5 concludes this paper and discusses action items for continuing to foster growth in this exciting interdisciplinary field.

## 80  2   AI and deep learning's applications in Arctic Sea ice research

### 2.1   Sea ice lead detection

Sea ice leads, which are distinct features of the polar seascape, are narrow, linear openings in sea ice that expose the ocean beneath (Tedesco, 2015). They vary in size, ranging from meters to several kilometers in width and can extend up to hundreds of kilometers in length. Their detection and monitoring are important due to their multifaceted role in both ecological and
climatological contexts. From an ecological perspective, these leads serve as vital habitats and migration paths for numerous Arctic species such as seals and polar bears, playing an essential role in sustaining polar biodiversity. From a climatological viewpoint, leads contribute significantly to the exchange of heat, moisture, and momentum between the ocean and atmosphere, influencing regional weather, sea ice dynamics, and global climate patterns. Additionally, accurate lead detection also aids in polar marine navigation, offering safer and more effective shipping routes.
The application of deep learning in sea ice lead detection primarily leverages satellite-based remote sensing data due to its comprehensive temporal and spatial coverage. Researchers employ a range of data types, from optical satellite imagery and synthetic aperture radar (SAR) data. In optical satellite imagery, thermal emissive bands are commonly used for lead detection, as leads exhibit higher temperatures relative to the surrounding ice due to their reduced albedo and increased solar absorption. Widely utilized instruments include MODIS and VIIRS (Hoffman et al., 2022, 2021; Qiu et al., 2023; Yin et al.,
2021). However, their relatively moderate spatial resolution (∼500 m) makes it difficult to use in identifying narrow leads. The recent advancements in satellite technology yield optical image products such as Landsat-8 (Yin et al., 2021), Sentinel-2 (Qiu et al., 2023), and SDGSAT-1 (Qiu et al., 2023), providing the spatial resolution (∼10-30 m) needed for narrow leads detection.

While optical satellite images offer extensive coverage and ease of access, data collection can only occur in clear sky conditions and in the presence of solar illumination. In comparison, SAR data can be obtained at fine spatial resolution and
can collect data in cloudy or dark conditions, yet being sensitive to surface roughness and susceptible to speckle noise. Notable examples of SAR data instruments encompass Sentinel-1 (Liang et al., 2022; Qiu et al., 2023; Zhong et al., 2023), RADARSAT-2 (Dawson et al., 2022) and HY-2B (Zhong et al., 2023). To enhance analytical capabilities, additional data types are generally also included in the analysis, including sea ice concentration (SIC) products from the National Snow and Ice Data Center



(NSIDC) for the ice pack mask (Hoffman et al., 2022), ERA5 meteorological data to provide climate variables that inform sea
ice conditions (Qiu et al., 2023; Yin et al., 2021), and CryoSat-2 to offer sea ice thickness data for additional context (Dawson
et al., 2022; Zhong et al., 2023).

The primary outputs of deep learning applications in sea ice lead detection are lead maps or masks that identify and delineate
sea ice leads from other ice types. Such maps or masks are usually generated at the same spatial resolution as the input data,
providing a pixel-wise classification where each pixel is labeled according to its identified class - 'lead', 'non-lead', or various
other ice types in the case of multi-class classification (Dawson et al., 2022; Hoffman et al., 2022; Liang et al., 2022). In
applications seeking to increase the spatial resolution of certain optical satellite data, the outputs can include high-resolution
reconstructions of these datasets. Deep learning techniques can be applied to upscale low-resolution images while preserving
important details and enhancing the visibility of fine-scale features such as sea ice leads (Yin et al., 2021). Another interesting
application of deep learning in sea ice lead detection involves network analysis (Kaltenborn et al., 2022). By modeling the
lead network as a graph, various network analysis methods can be applied to understand its temporal change (e.g., shrinking
or expanding).

For deep learning methods, semantic segmentation is the method of choice in many sea ice lead detection studies because of
its ability to classify each pixel in an image, thereby providing a detailed and comprehensive representation of the leads amidst
the large ice cover. In terms of model selection, U-Net has become a popular choice due to its encoder-decoder structure (Hoff-
man et al., 2022; Ronneberger et al., 2015). The encoder part of U-Net captures the spatial context in the image, while the
decoder part enables precise localization for delineating the boundaries of leads. This architecture also includes skip connec-
tions, which help in recovering the fine-grained details that might be lost during the downsampling in the encoder part. Dulam
et al. (2022) applied the U-Net architecture, which takes multiple modality data (e.g., visible-band WorldView and C-band
Sentinel) as input and performed accurate lead predictions with a small amount of training data. Liang et al. (2022) developed
an entropy-weighted network (EW-Net) by improving the baseline U-Net with additional modules, such as dense blocks. Dense
blocks, originating from DenseNet (Li and Hsu, 2020), strengthen feature propagation and increase feature reuse, leading to
improved gradient flow and model compactness. With these improvements, EW-Net has demonstrated better model predictive
performance than the baseline U-Net model.

In addition to U-Net, researchers have explored other techniques to separate leads from ice floes, including both traditional
machine learning approaches such as thresholding (Qiu et al., 2023; Zhong et al., 2023), the waveform mixture algorithm
(Zhong et al., 2023), and deep learning models such as DeepLabv3 (Yurtkulu et al., 2019). Dawson et al. (2022) formulated
lead detection as a classification problem and employed 1D CNN to classify leads at the sampled data points. The authors
compared their approach with a decision-tree method and found that the CNN-based approach led to fewer false positives and,
therefore, higher classification accuracy. For data preprocessing, such as super resolution, a model may include a dedicated
branch for upscaling the input data (Yin et al., 2021). Table 1 summarizes the main deep learning techniques and methods to
support sea ice lead detection.



**Table 1.** Deep learning solution techniques for sea ice lead detection

| Sea ice application | Deep learning problem formulated | Deep learning techniques (models) | Output | References |
|---|---|---|---|---|
| Sea ice lead detection <br>  <br> Source: NASA | Multi-class, pixel-wise classification | Semantic Segmentation (U-Net, DeepLab, EW-Net) | Lead maps or masks that delineate sea ice leads from non-lead area | Dulam et al. 2022; Hoffman et al. 2022; Liang et al. 2022; Ronneberger et al. 2015; Yurtkulu et al. 2019 |
| | Network analysis | Graph Convolutional Network (EvolveGCN) | Statistics to reveal the change (e.g., expansion or shrinking) in the sea ice lead network | Kaltenborn et al. 2022 |
| | Point-based classification | Convolutional Neural Network (1D CNN) | Point type classification (such as lead or ice floe) at a certain sampled location | Dawson et al. 2022 |

## 2.2 Sea ice thickness estimation

Sea ice thickness (SIT) estimation refers to the measurement and calculation of sea ice depth. The thickness modulates the exchange of heat, moisture, and gas between the atmosphere and the ocean, thereby exerting significant effects on oceanic

circulation, sea level, and both marine and terrestrial ecosystems. Furthermore, the depth of sea ice is also an important climate change indicator. In the context of global warming, the Arctic and Antarctic sea ice have been undergoing significant changes, with sea ice thickness generally decreasing over the years (Meier, 2017; Perovich and Richter-Menge, 2009). Therefore, accurate and efficient methods for estimating sea ice thickness are essential to monitor and predict these changes, enabling scientists to better understand the ongoing alterations in our climate system and their potential impacts.

Within the realm of sea ice thickness estimation using deep learning, various types of data can be used. The brightness temperature data sourced from AMSR2 (Chi and Kim, 2021; Dong et al., 2022) provide a primary dataset, measuring the radiation naturally emitted by the sea ice within the microwave spectrum and exhibiting a strong correlation with sea ice thickness. The advantage of AMSR2 data is its ability to capture information under all weather conditions and during both day and night, offering continuous and reliable coverage. Additional input data encompass satellite products, including optical

(Dawson et al., 2022) and SAR (Dawson et al., 2022; Shamshiri et al., 2022), capturing surface features of sea ice and indirectly inferring its thickness. Furthermore, thermodynamic parameters such as air temperature, sea surface temperature, wind speed,





and snow depth, derived from reanalysis data such as ERA5 (Liang et al., 2023; Liu et al., 2023), can suggest sea ice thickness indirectly, reflecting the conditions conducive to ice formation and evolution. The strength of reanalysis data lies in their global coverage and continuity, but they are model-derived and not direct measurements, so they may contain biases or uncertainties.

To train deep learning models effectively, ground truth data is indispensable for learning the correlation between input data and sea ice thickness. Ground truth data can be obtained from in-situ measurements such as AWI Moored Upward Looking Sonar (ULS) data (Shamshiri et al., 2022) or calculated from CryoSat-2 data (Chi and Kim, 2021; Dawson et al., 2022; Liang et al., 2023), which are explicitly designed to monitor polar ice thickness fluctuations and variations in sea ice. Numerical simulation models like Pan-Arctic Ice Ocean Modeling and Assimilation System (PIOMAS; Zhang and Rothrock, 2003)

provide another source of sea ice thickness data, despite potential limitations and uncertainties (Liu et al., 2023). In terms of output, deep learning models typically generate a continuous map of sea ice thickness, usually in the form of gridded datasets representing the estimated sea ice thickness for specific geographic locations. The resolution of these datasets is contingent on the input satellite data and deep learning model specifics. These datasets then serve as foundations for subsequent analysis and climate model input.

Sea ice thickness estimation is typically addressed as a regression problem, given the output's continuous nature. The goal is to predict a continuous value - the sea ice thickness, using the input data. Traditional machine learning models have been widely utilized for this purpose (Glissenaar et al., 2023; Herbert et al., 2021; Shamshiri et al., 2022; Yan and Huang, 2020; Zhao et al., 2023). For instance, Shamshiri et al. (2022) applied random forests regression to estimate sea ice thickness, leveraging input data features such as backscatter intensity, texture, and polarization from Sentinel-1. Similarly, CryoSat-2's radar altimeter

readings can be converted to ice thickness, aided by knowledge of ice and snow densities (Glissenaar et al., 2023). Random forest regression was applied in this study as well.

     In recent years, there has been a shift towards the use of deep learning models for sea ice thickness estimation because of their enhanced capacity to model intricate relationships, handle extensive high-dimensional data, and learn useful features autonomously (Landy et al., 2022). A typical solution is to formulate the problem as regression analysis. Herbert et al. (2021)

applied a regression-based fully connected neural network to estimate SIT from multiple input features, such as brightness temperature, surface temperature, and sea ice concentration. Yan and Huang (2020) developed a 1D CNN to fuse image features and create sea ice thickness products from multi-source satellite data, such as AMSR2 and Soil Moisture Ocean Salinity (SMOS) dataset. Similarly, Chi and Kim (2021) advanced the model by developing ensembled 1D CNN for SIT estimation, and the results have shown to achieve high consistency with reference data. Liang et al. (2023) developed a self-

attention CNN to incorporate thermodynamic parameters to estimate the daily SIT in the Arctic winter. The self-attention module was employed to explore the hidden connections among the extracted features to achieve a more accurate estimation.

     SIT estimation was also addressed as a clustering problem. Moreau et al. (2023) integrated deep learning with Bayesian inference to estimate sea ice thickness by analyzing icequake waveforms. A deep scattering network was applied to convert the original data space to an abstract space where feature similarity can be better represented. These abstracted features are

then clustered to create sub-families of icequake waveforms. Based on this result, Bayesian inversion was further applied to infer the ice thickness. SIT estimation can also be cast as a forecasting task, using temporal models such as PredRNN++ to





**Table 2.** Deep learning solution techniques for sea ice thickness estimation

| Sea ice application | Deep learning problem formulated | Deep learning techniques (models) | Output | References |
|---|---|---|---|---|
| Sea ice thickness estimation  Source: NSIDC | Regression | Regression fully connected neural network | Thematic maps showing sea ice thickness | Herbert et al. 2021 |
| | | Convolutional Neural Network (1D CNN, Attention-based CNN) | | Chi and Kim 2021; Liang et al. 2023; Yan and Huang 2020 |
| | Clustering | Deep scattering network | | Moreau et al. 2023 |
| | Time-series forecasting | Recurrent Neural Network (PredRNN++) | Near-term forecast of sea ice thickness, creating continuous sea ice thickness maps in future timestamps | Liu et al. 2023 |

predict future sea ice thickness (Liu et al., 2023). By introducing a spatiotemporal memory flow, PredRNN (Wang et al., 2022d) is capable of capturing the decoupled long- and short-term dynamics and the spatiotemporal variations in the sequence data. Table 2 summarizes the main deep learning techniques and methods to support sea ice thickness estimation.

## 2.3 Sea ice concentration (SIC) classification

Sea Ice Concentration (SIC) is a measure of the proportion of sea area covered by ice, usually expressed as a percentage. The process of assigning categories to specific levels of ice coverage in a given sea area is known as SIC classification. For climate scientists, SIC classification is a key metric for understanding the impacts of global warming, as it directly affects the Earth's energy balance. For the shipping industry, accurate predictions of SIC can enhance navigation safety and efficiency in ice-infested waters. Thus, timely and accurate SIC classification is of great importance.

Datasets from AMSR2, RADARSAT2, and Sentinel-1 are frequently leveraged for SIC classification via deep learning algorithms, given their unique capabilities and strengths. AMSR2 offers brightness temperature data, which is highly relevant for analyzing ice concentration (Chi et al., 2019; Feng et al., 2023; Liu et al., 2022a; Malmgren-Hansen et al., 2021). As already mentioned, the advantage of AMSR2 lies in its cloud penetration capacity, which ensures continuous data acquisition. Additionally, AMSR2's multi-frequency operation facilitates the recording of varying ice properties. However, its low spatial resolution limits the detail of ice structure that can be captured. RADARSAT2, on the other hand, delivers SAR data, distinguished by its high spatial resolution, which allows for capturing detailed ice feature. In addition, its operational independence



from sunlight or atmospheric conditions makes it suitable for polar regions monitoring (Cooke and Scott, 2019; Nagi et al., 2020; Sola et al., 2020). Nevertheless, RADARSAT2 data processing is complicated due to the complex nature of SAR signals,

including speckle noise. In comparison, the Sentinel-1 mission offers C-band SAR data, which incorporates the strengths of SAR data and some unique benefits. C-band SAR data is sensitive to sea ice changes and is effective in detecting thin ice and open water areas (Dominicus et al., 2021; Malmgren-Hansen et al., 2021; Stokholm et al., 2022; Wang and Li, 2021b). The Sentinel-1 satellites also provide a high revisit frequency, ensuring updated and timely data. Like RADARSAT2, it also requires advanced processing and interpretation of SAR signals.

The ground truth data utilized for training deep learning models for SIC classification stem from various sources, each bearing unique strengths and weaknesses. For instance, expert-crafted ice charts, such as those from the Greenland Ice Service, serve as invaluable data repositories. These charts incorporate a wealth of human experience and understanding of sea ice, providing context often absent in raw satellite data. However, they are subject to subjective interpretation and have limited frequency and coverage. Alternatively, satellite-derived products from the MODIS or AMSR-E are beneficial due to their ex-

pansive coverage, high temporal frequency, and objectivity. Nonetheless, they can be affected by cloud cover, and the accuracy can be contingent on the ice concentration calculation algorithms. The choice of ground truth data is guided by the specific study requirements, and a combination of different sources may be employed to optimize their strengths and mitigate their weaknesses.

Deep learning-based SIC classification from satellite data can be treated as either a regression or a classification problem, de-

pending on the task formulation. When viewed as a regression problem, the aim is to predict a continuous variable that indicates the sea ice concentration (Chi et al., 2019; Cooke and Scott, 2019). Despite providing granular sea ice concentration estimates, this method can be more challenging due to the complexity of modeling continuous variables. Alternatively, SIC estimation as a classification problem entails predicting discrete classes like "no ice," "partial ice," and "full ice" (Nagi et al., 2020; Sola et al., 2020), simplifying the problem and potentially offering a more robust model, particularly with limited or noisy data.

Semantic segmentation models are commonly used to achieve this goal. For example, U-Net, which is designed specifically for image segmentation, is frequently employed (Dominicus et al., 2021; Stokholm et al., 2022; Wang and Li, 2021b). Another segmentation model, DeepLab, which uses dilated convolutions to enable multi-scale contextual information aggregation without resolution loss, also shows good performance for the SIC estimation task (Feng et al., 2023; Malmgren-Hansen et al., 2021). Super-resolution techniques applied to satellite data enhance SIC classification (Feng et al., 2023; Liu et al., 2022a) by

improving the spatial resolution of data, allowing for the capture of finer ice structure details, thereby potentially improving the SIC classification accuracy.

SIC concentration can also be formulated as a regression model predicting a continuous SIC value (Chi et al., 2019; Cooke and Scott, 2019; Nagi et al., 2020). Chi et al. (2019) developed a multilayer perceptron (MLP) deep learning-based regression model to predict the SIC from AMSR2 passive microwave data. The spectral unmixing algorithm was first applied to higher-

resolution MODIS imagery to better discern first-year ice from melt ponds, as their spectral signals are often mixed within a single image pixel. This sub-pixel level analysis yields an accurate ice/water classification fractional abundance map. This class information was further mapped to a lower-resolution AMSR2 image to retrieve the percentage of sea ice cover within



**Table 3.** Deep learning solution techniques for sea ice concentration estimation

| Sea ice application | Deep learning problem formulated | Deep learning techniques (models) | Output | References |
|---|---|---|---|---|
| Sea ice concentration (SIC) estimation | Multi-class, pixel-wise classification | Semantic segmentation (U-Net, DeepLab) | Thematic map indicating SIC types, e.g., "no ice," "partial ice," and "full ice" for each image patch | Dominicus et al. 2021; Feng et al. 2023; Malmgren-Hansen et al. 2021; Stokholm et al. 2022; Wang and Li 2021b |
| Source: NSIDC | Regression | MLP, CNN (AlexNet, VGG, ResNet, DenseNet) | Predict a continuous variable indicating SIC | Chi et al. 2019; Cooke and Scott 2019; Nagi et al. 2020, 2021 |

the image pixel. Based on this data annotation mechanism, the MLP was further trained to predict the SIC values. Results have shown better regional and pan-Arctic accuracy than other SIC products. Cooke and Scott (2019) applied SAR images and

CNN-based regression to derive SIC. Instead of using ice charts as the ground truth, the authors adopted passive microwave data for the estimation of SIC using the ARTIST Sea Ice algorithm. In addition, patch-based analysis was applied, allowing the model to incorporate more spatial contextual information in its learning process. Other similar works include Nagi et al. (2020). In these works, commonly used CNN architectures include fewer-layer AlexNet and VGG and deeper models such as DenseNet and ResNet (Li and Hsu, 2020). Table 3 summarizes the main deep learning techniques and methods to support sea

ice concentration estimation.

## 2.4 Forecasting the change in sea ice extent

Sea Ice Extent (SIT) forecasting is a process where advanced methods and models are used to predict the level of sea ice in a specific region over a future period. SIT is defined as the area of the ocean covered by more than 15% of SIC. Hence, SIT forecasting can be treated as an SIC forecasting problem, and based on the result, we can further compute the SIT. These

forecasts are important for several reasons. First, they are vital for ship navigation in polar regions, where unexpected changes in ice conditions can pose significant hazards. Second, they play a critical role in climate science as sea ice influences the heat balance of the Earth. Third, these forecasts contribute to the understanding of the impacts of climate change, as declining sea ice is a key indicator of global warming.

Several sources of data contribute to SIT forecasting. The NSIDC, NASA, and the Japan Aerospace Exploration Agency

(JAXA) all provide SIC data through algorithm-based estimation from multiple satellite products (Grigoryev et al., 2022; Kim et al., 2020; Ren et al., 2022; Zhang et al., 2022a). These datasets are very helpful due to their large spatial and temporal



coverage. However, their accuracy can be affected by atmospheric conditions and surface melt. Furthermore, discrepancies may arise due to the use of varying sensors, processing algorithms, or calibration methodologies. Reanalysis data incorporates ERA5 and ERA-Interim outputs from the European Centre for Medium-Range Weather Forecasts (ECMWF), creating a coherent

long-term record of numerous atmospheric, terrestrial, and oceanic phenomena. Additionally, the Ocean Reanalysis and derived ocean heat content (ORAS4) dataset, another ECMWF product, offers insights into oceanic temperature, salinity, currents, and sea levels. Reanalysis datasets complement SIC forecasts by providing data on atmospheric and oceanic factors influencing sea ice dynamics, including air temperature, wind speed, ocean currents, and sea surface temperature (Asadi et al., 2022; Chi et al., 2021; Liu et al., 2021a, c). Ice charts from the Canadian Ice Service (CIS) provide expert-verified information about ice

conditions in Canadian waters. They contain information about ice type, concentration, thickness, and stage of development. While they do contain SIC data, it is often in a categorical rather than a continuous format (e.g., "9/10 to 10/10" rather than "95%"). These ice charts can be used as complementary data for SIC forecasting, providing high-accuracy, ground-truth data for model validation or data assimilation (Asadi et al., 2022). However, their spatial and temporal coverage is more limited than the satellite or reanalysis data.

Deep learning models used for SIC forecasting exhibit common characteristics. The input typically comprises a sequence of historical data points, including past SIC data and relevant atmospheric and oceanic conditions. Preprocessing and normalization of the gridded data are necessary, along with handling missing values through techniques such as interpolation. The output prediction for a future time frame mirrors the input data format, facilitating comparison and evaluation. U-Net, LSTM, and Convolutional Long Short-Term Memory (ConvLSTM) models are commonly used deep learning architectures for SIC

forecasting.

U-Net's structure allows for both high and low-level feature capture, and its symmetric expansion path helps regain spatial information lost during the contracting phase, aiding in precise localization. However, modifications to the architecture or data representation are often required as U-Net does not inherently manage temporal dependencies (Andersson et al., 2021; Grigoryev et al., 2022). Optimizations on the U-Net include the development of SICNet (Ren et al., 2022), a U-Net architecture

with a temporal-spatial attention module to capture the spatiotemporal dependencies within the input data. Note that the use of U-Net is a surrogate for a forecasting task. Even though the model captures temporal relationships within the data, the problem remains solved as a spatial problem. It is not a sequence-to-sequence learning method commonly adopted in time series prediction.

Recurrent neural network (RNN) models are often employed in forecasting tasks due to their sequence-to-sequence learning

ability. Choi et al. (2019) applied a gated RNN to provide a 15-day prediction based on historical SIC data. Chi and Kim (2017) developed an LSTM model that also uses SIC sequences for future one-month prediction of SIC. Wei et al. (2022) developed an attention-based LSTM neural network model that considers the contribution from all hidden states in the forecasting. The results show that the proposed model outperforms traditional LSTM, which only considers information provided at the last hidden state. Similar works also include Ali et al. (2021). However, LSTM-based models capture only the temporal correlation,

largely ignoring the spatial relationships. To address this issue, ConvLSTM, an LSTM variant that efficiently handles temporal dependencies by incorporating convolutions within the LSTM cell, allows for the simultaneous processing of both spatial and





**Table 4.** Deep learning solution techniques for sea ice concentration estimation

| Sea ice application | Deep learning problem formulated | Deep learning techniques (models) | Output | References |
|---|---|---|---|---|
| Sea ice extent forecasting <br><br> Source: NSIDC | Time-series forecasting | Semantic segmentation (U-Net, SICNet) | Classify each pixel into a SIC type, with multi-channel output and each channel represents the SIC classification for a future timestamp (e.g., month) | Andersson et al. 2021; Grigoryev et al. 2022; Ren et al. 2022 |
| | | RNN (LSTM, attention-based LSTM, gated RNN) | Predict sea ice concentration in future timestamps through sequence-to-sequence learning. The forecast could be for one or multiple timestamps | Ali et al. 2021; Chi and Kim 2017; Choi et al. 2019; Wei et al. 2022 |
| | | CNN+RNN (ConvLSTM, multi-task ConvLSTM) | | Chi et al. 2021; Kim et al. 2021; Liu et al. 2021a, b; Zhang et al. 2022a |

temporal data. Despite their resource-intensive nature, ConvLSTM models are particularly effective for spatiotemporal tasks like SIC forecasting (Liu et al., 2021a, b). Optimizations on these models include a two-stream ConvLSTM to learn the sparse and detailed characteristics of sea ice dynamics (Chi et al., 2021), a multi-task ConvLSTM for simultaneous SIC and sea ice

extent (SIE) prediction (Kim et al., 2021), and the addition of a Conditional Random Field (CRF) module for boundary and edge detection (Zhang et al., 2022a). Table 4 summarizes the main deep learning techniques and methods to support sea ice extent forecasting.

## 2.5 Sea ice motion estimation

Sea ice motion estimation refers to the assessment of sea ice displacement and velocity over time. Such movements, influenced

by wind, ocean currents, and temperature variations, can have profound effects on the polar marine ecosystem, climate change, and human activities. Sea ice motion contributes to the redistribution of ice thickness and coverage, therefore impacting the Earth's albedo effect. The movement and subsequent melting of ice in warmer regions contributes to the global thermohaline





circulation, which significantly influences climate patterns. For human activities, understanding sea ice motion is critical for
safer navigation routes for shipping and other maritime operations in the Arctic regions. Therefore, accurate sea ice motion
estimation is important in both scientific and socioeconomic contexts.

Data used in sea ice motion estimation comes from a variety of remote sensing platforms. For instance, passive microwave
sensors such as AMSR-E and AMSR2 are useful in sea ice motion estimation as they can detect ice concentration and type,
and are unaffected by darkness or cloud cover, enabling continuous monitoring (Liu et al., 2022a; Petrou et al., 2018; Petrou
and Tian, 2017, 2019). However, these sensors' relatively coarser spatial resolution poses a challenge for detailed local studies.
In contrast, SAR imagery from sources such as Sentinel-1 provides high-resolution data, enabling the detection of smaller-
scale ice movements (Petrou et al., 2018). The technology also operates in all weather conditions and during both day and
night. However, the interpretation of SAR images can be complex due to the influences of surface roughness and moisture
content. Additionally, complementary data from sources like ASCAT (Advanced Scatterometer), which measures radar sig-
nal backscatter, and reanalysis data from Japanese 55-year Reanalysis (JRA55), can enhance the sea ice motion estimation
process by providing surface wind data. Wind plays a significant role in driving sea ice motion, thus data from ASCAT or
JRA55 enhances the estimation process (Petrou and Tian, 2019; Zhai and Bitz, 2021). However, ASCAT measurements can be
affected by heavy rain, and JRA55 data, based on model simulations and assimilated observational data, may contain biases
and inaccuracies. Sea ice velocity data from NSIDC offers direct measurements of sea ice motion and can act as ground truth
for model validation (Zhai and Bitz, 2021), although its coarser spatial and temporal resolution and potential errors due to the
mix of satellite and buoy data may pose challenges.

The output of sea ice motion estimation often takes the form of two-dimensional maps or vectors showing the displacement
over time. The vectors typically indicate the direction and speed of the ice movement, often provided in grid format or as
displacement fields. These displacement maps offer a visual and quantifiable representation of sea ice movement, facilitating
the understanding of ice dynamics and interactions with atmospheric and oceanic forces.

There are generally two ways for sea ice motion detection. The first is to predict a future image, and then, by identifying the
same sea ice patches on the pair of images in consecutive timestamps, one can create the motion vectors for each pixel within
the image scene. This is commonly achieved through video prediction techniques in an unsupervised learning framework
(Lotter et al., 2016). Srivastava et al. (2015) developed an encoder-decoder-based LSTM model in which the encoder LSTM
transfers the input into a 1D representation. Then, multiple decoder LSTMs are introduced to reconstruct images from the
representation, learning to predict images in the future timestamps. The second approach is to first calculate optical flow,
which indicates sea ice motion from the pair of images, creating a two-dimensional displacement field that symbolizes sea ice
motion. Such optical flow can be incorporated into LSTM or ConvLSTM models for future motion prediction, leveraging these
models' ability to capture sequential data's temporal dependencies (Petrou and Tian, 2017, 2019).

The difference between LSTM and ConvLSTM is that LSTM takes in 1D sequence data, so the sea ice motion image
will need to be serialized before applying it. ConvLSTM was able to add a convolutional module to an LSTM and has the
ability to process 2D images in a time sequence. In another method, inputs including surface wind data, the previous day's
sea ice velocity, and Sea Ice Concentration (SIC) feed into a CNN, subsequently engaging in a patch-wise regression task





**Table 5.** Deep learning solution techniques for sea ice concentration estimation

| Sea ice application | Deep learning problem formulated | Deep learning techniques (models) | Output | References |
|---|---|---|---|---|
| Sea ice extent forecasting | Video prediction | Unsupervised encoder-decoder LSTM | Predict the future sea ice images and generate the motion vector by comparing a pair of images in consecutive timestamps | Srivastava et al. 2015 |
| Source: measuring sea ice motion | Point-based Regression | CNN + fully connected layers | Predict the motion vector at both x and y directions for each pixel within the study area | Zhai and Bitz 2021 |
| | Time-series forecasting | LSTM (1D), Convolutional LSTM (2D) | | Petrou and Tian 2017, 2019 |

to predict sea ice velocity (Zhai and Bitz, 2021). This technique effectively utilizes environmental factors' interaction and their impact on sea ice motion by allowing the CNN to extract complex spatial features from the input data. The results

also demonstrated the superior performance of the CNN-based regression model than other baseline models, such as linear regression and random forest. This is owing to the CNN's ability to capture spatial relationships among the pixels within an image scene, whereas in other baseline models, each pixel is considered an independent sample. Recently, some studies have adopted super-resolution techniques to enhance satellite imagery resolution prior to optical flow computation (Liu et al., 2022a; Petrou et al., 2018), enabling a more detailed representation of sea ice, hence improving the optical flow and the overall motion

prediction's accuracy. Table 5 summarizes the main deep learning techniques and methods to support sea ice motion estimation.

## 2.6 Sea ice type classification

Sea ice type classification is research aimed at distinguishing between various types of sea ice, such as multi-year ice, first-year ice, new ice, and ice-free areas (e.g., Tedesco, 2015). These classifications aid in understanding climate change and the functioning of polar ecosystems, as different types of ice have varying thermal properties, saline contents, and associated mi-

croorganism populations. Moreover, the presence of certain ice types can pose risks to shipping routes and offshore structures. With the increasing accessibility of the Arctic region due to ice melting, accurate sea ice type classification is becoming more important for safe navigation and efficient resource extraction.



Sea ice type classification utilizes a variety of data sources, each with distinct characteristics and applications. The data sources can be broadly categorized into satellite imagery and aerial photographs. Satellite data, particularly SAR systems including Sentinel-1, RADARSAT-2, and Gaofen-3, hold substantial value due to their all-weather and diurnal observation capabilities. Sentinel-1 provides medium-resolution images in dual polarization, crucial for ice type differentiation through backscattering coefficients (Boulze et al., 2020; Song et al., 2018, 2021). RADARSAT-2 provides flexible beam modes and polarizations, making it a versatile tool for capturing different scales of ice features (Chen et al., 2023; Jiang et al., 2022a, b). Meanwhile, Gaofen-3, a Chinese SAR satellite, combines high-resolution imaging and a large swath width, enabling detailed large-scale ice surveys (Zhang et al., 2022b). In contrast, the multispectral imaging satellite Landsat-8 provides high spatial resolution visible and infrared images. It is useful for ice type differentiation due to its ability to capture distinct spectral characteristics of different ice types (Cáceres et al., 2022; Han et al., 2020). However, unlike SAR systems, their usage is limited by cloud cover and polar darkness. Aerial photographs, while offering extremely high-resolution data, capture detailed ice structures and features but are resource and time-intensive with relatively small coverage (Sudakow et al., 2022).

Verification and training of machine learning models in sea ice type classification often rely on expert-annotated ice charts and researcher-led manual annotations. Ice charts, such as those from the Canadian Ice Service (CIS) and the Ice Service department of the German Federal Maritime and Hydrographic Agency (BSH), constitute human interpretation of ice conditions based on multiple sources, including aerial surveys, ship reports, and satellite data. Despite their use as reliable sources for large-scale, temporal studies (Boulze et al., 2020; Cáceres et al., 2022), the spatial resolution of these charts is generally coarse and may not cover all regions or times of interest. Additionally, inconsistencies might arise from differential interpretations of ice conditions across analysts and institutions. Manual annotation offers another method for establishing ground truth, involving detailed examination of high-resolution imagery and annotating each pixel or region with the appropriate ice type. Although this yields high-accuracy and high-resolution labels suitable for model training and validation (Han et al., 2020; Sudakow et al., 2022), it is a laborious process requiring substantial expertise in ice analysis, and feasibility depends on the availability of high-quality images and accessibility to ice areas.

The output from sea ice type classification is typically presented as thematic maps, indicating the geographical distribution of various ice types. This format facilitates a visual and spatial understanding of the ice conditions. The color-coded maps usually represent different types of sea ice and open water, along with metadata providing information about data source, time, and possibly uncertainties. Outputs may also be represented numerically, useful for climate models and further computational analysis.

The field of sea ice type classification utilizes both traditional machine learning techniques and more recent deep learning models. Traditional models, such as SVMs and MLPs, are commonly employed for patch-wise classification (Cáceres et al., 2022; Han et al., 2020). Despite their simplicity and computational efficiency, they may struggle with complex and high-dimensional data. In contrast, deep learning models, such as CNNs, can learn useful features directly from raw data. ResNet has been widely used for patch-wise classification due to its strong capability in handling image data (Boulze et al., 2020; Liu et al., 2022b; Pedersen and Kim, 2020; Song et al., 2018). Advanced architectures such as Multiscale MobileNet (Zhang et al.,



2022b) and Normalizer-free ResNet (Lyu et al., 2022b, a) further improve performance by introducing efficient computation and improved training dynamics.

Some researchers approach the problem as a semantic segmentation task, employing models like U-Net or advanced architectures like the Swin Transformer-based U-Net. The Transformer is a new architecture with the capability of capturing data dependencies over a long range, thereby possessing stronger feature extraction capability than other feature extraction backbones. These models are also capable of generating high-resolution classification maps and are often better at preserving spatial structures (Sudakow et al., 2022). Semi-supervised methods, such as an initial over-segmentation using Iterative Region Growing using Semantics (IRGS) segmentation, followed by ResNet-based patch classification, offer an interesting alternative (Jiang et al., 2022a, b). This approach can improve the handling of boundaries and heterogeneous areas but might require careful parameter tuning and a thorough understanding of the data. Some researchers also incorporate temporal information, leveraging ConvLSTM networks (Song et al., 2021). This method can account for temporal variations in ice conditions but adds an extra layer of complexity and computational demand. In summary, the selection of the appropriate method largely depends on the specific requirements of the study, available resources, and the nature of the data at hand. Table 6 summarizes the main deep learning techniques and methods to support sea ice type classification.

## 3 Deep learning strategies tailored for Arctic Sea ice research

Besides adopting different types of deep learning models, customized learning strategies, such as super-resolution and refined loss functions, have also been developed to support Arctic sea ice research. These advances bring new depth and accuracy to data analysis and forecasting tasks. The following section further explores and discusses these strategies.

### 3.1 Super-resolution

In the domain of image processing, super-resolution refers to a set of methodologies that augment the spatial resolution of an imaging system. In the deep learning realm, super-resolution techniques employ trained neural networks to recreate high-resolution depictions from low-resolution equivalents, thereby filling in the details that the initial imaging process was unable to document (Dong et al., 2014).

For example, in the application of sea ice lead detection, high-resolution imagery is critical to ensure accurate lead delineation because of the typically small and narrow shape of these features. Satellite imagery, such as MODIS, carries inherent constraints in spatial resolution, making it less suitable for detecting smaller leads. As a solution to this, Yin et al. (2021) proposed a model named DeepSTHF (deep sub-pixel turbulent heat flux), which employs MODIS Ice Surface Temperature (IST) images as inputs to generate enhanced output images and lead maps. The proposed architecture achieves this using two separate CNN branches. The first branch, composed of residual blocks, is committed to refining the MODIS IST image resolution, enabling complex image detail learning while curtailing information loss during the enhancement process. In parallel, the second branch, established on an encoder-decoder framework, is tasked with the generation of a high-resolution lead map.



**Table 6.** Deep learning solution techniques for sea ice type classification

| Sea ice application | Deep learning problem formulated | Deep learning techniques (models) | Output | References |
|---|---|---|---|---|
| Sea ice type classification <br><br> Perovich et al. 2020 | Patch-based classification | CNN (ResNet, Multi-scale MobileNet, Normalizer-free ResNet) | Thematic map indicating sea ice types, e.g., "multi-year ice," "first-year ice," and "new ice" for each image *patch* | Boulze et al. 2020; Liu et al. 2022b; Lyu et al. 2022b; Pedersen and Kim 2020; Song et al. 2018; Zhang et al. 2022b; |
| | Semi-supervised learning | Unsupervised segmentation + patch-level classification (IRGS+CNN) | | Jiang et al. 2022a, b |
| | Pixel-based segmentation | Semantic segmentation (U-Net, Swin Transformer-based U-Net) | Thematic map indicating sea ice types, e.g., "multi-year ice," "first-year ice," and "new ice" for each image *pixel* | Sudakow et al. 2022 |
| | Time-sequence analysis | ConvLSTM | Classify sea ice type based on past sea ice conditions | Song et al. 2021 |

In the context of SIC classification, continuous SIC monitoring largely depends on passive microwave images. However, due to the coarse spatial resolution of these images, SIC identification tends to be blurred at the ice-water boundaries. Addressing this issue, works by Liu et al. (2022a) and Feng et al. (2023) applied the PMDRnet (progressive multiscale deformable residual network) to amplify the spatial resolution of sea ice passive microwave images based on unique characteristics of the images and sea ice movements. The network employs a strategy of multi-image super-resolution, with image alignment facilitated by a progressive alignment approach that dynamically switches reference images during the alignment of two neighboring images. This alignment is anchored on the principles of deformable convolution (DConv) and dilated convolution. DConv offers an augmentation of spatial sampling locations through learned offsets, thereby enhancing the geometric transformation modeling capacity of original CNNs that are constrained by a fixed kernel size in a convolutional layer. Furthermore, DConv





alignment utilizes multiple offsets at each feature location, enabling information capture in local neighborhoods and mutual complementing, thus resulting in increased alignment accuracy.

Super-resolution has also been applied in sea ice motion research. Petrou et al. (2018) employed super-resolved satellite images to derive optical flow and, therefore, a dense motion vector field with continuous values and subpixel precision, consequently achieving improved performance than using images at its original resolution. Super resolution can also be supported by incorporating temporal information. Liu et al. (2022a) leveraged temporal data to boost the resolution of coarse passive microwave data in their study. They employed a deep learning-based multi-image super-resolution approach to classify SIC. Here, the temporal attention mechanism helps extract and fuse temporal information across the data sequence, effectively compensating for any image alignment issues, thereby ensuring a more accurate and fine-grained SIC classification.

## 3.2 Customizing loss function in AI models

A loss function, in the context of deep learning, is a measure of how far the prediction of a model is from the true output (a.k.a., ground truth). It quantifies the error or discrepancy between the predicted and actual values, offering a way to evaluate the performance of a model. In a typical deep learning framework, the model is trained by adjusting its parameters iteratively to minimize the loss function. Different loss functions have different mathematical properties and are suitable for different types of prediction tasks. For instance, Mean Squared Error (MSE) is often used for regression tasks, while Cross-Entropy loss is commonly used for classification tasks. Sometimes, however, standard loss functions may not adequately capture the specific objectives or nuances of a certain task. In such situations, researchers may develop custom loss functions tailored to the problem at hand, as evident in the following examples.

In the realm of sea ice type classification, certain ice types are naturally less commonly seen, and consequently, a model trained on such data may tend not to predict these rare types. Pedersen and Kim (2020), acknowledging this class imbalance, proposed a true negative weighted loss. This method reduces the loss value when the model correctly identifies the absence (a true negative) of a specific type of ice. Consequently, for ice types that are rare throughout the dataset, the model is not disincentivized to predict their presence. This approach ensures that the model's predictions are largely influenced by the image content, as opposed to the overall statistical distribution of different ice types.

A different problem arises in SIC classification, where passive microwave radar satellite data is susceptible to interference from atmospheric variations, and calibration for one area may not translate well to another. To address this issue, Dominicus et al. (2021) proposed a loss function that accounts for the uncertainty in passive measurements, incorporating uncertainty from a SIC product (e.g., OSI-401-b) into the loss computation. This uncertainty-weighed loss function scales the loss values by the uncertainty level, thereby preventing over-penalization of the model in instances where the concentration label exhibits high uncertainty. Similarly, Liu et al. (2022a) provided another example of tailoring loss functions to consider data characteristics. In addition to conventional reconstruction error for super-resolution, they proposed a SIC-specific loss function to enrich the model with more sea ice details for finer SIC predictions. The loss function leverages the polarization difference of brightness temperatures at the 89-GHz channel, a parameter frequently utilized for SIC computations. By assessing the difference in



this value between the reconstructed and ground truth images, the loss function encourages the reconstructed image to retain equivalent SIC information as the ground truth.

     In SIC forecasting, this task entails a sequence of SIC inputs coupled with a series of SIC predictions. Traditional loss functions, such as Mean Squared Error (MSE) or Mean Absolute Error (MAE), are commonly employed to measure the divergence between each forecast-ground truth pair (Wei et al., 2022). However, this approach fails to acknowledge the temporal

dependencies that characterize the underlying SIC trend, as SIC variations are not independent occurrences but are part of a sequential continuum. Addressing this gap, Liu et al. (2021b) proposed a gradient loss function (Grad-loss), designed to capture the SIC trend, thereby empowering the model to capture the temporal evolution of ice concentration. Similarly, Chi et al. (2021) utilized a perceptual loss function that takes spatial patterns into account, thereby capturing pixel interrelations more effectively than pixel-level loss. This method involves passing both predicted and ground truth images through a CNN and

computing loss based on the resultant feature maps. Moreover, SIC prediction regions may include land areas where sea ice cannot potentially form. Integrating predictions of these areas into loss computations may result in image content imbalance if the land cover area is extensive, potentially causing the model to skew towards predicting lower SIC values. Addressing this challenge, Kim et al. (2021) and Andersson et al. (2021) applied a land mask on SIC predictions to eliminate land regions from the loss computation, allowing for more efficient model optimization on grid cells that are feasible for ice formation.

## 3.3    New learning strategies to address for the lack of ground truth data

     As a data-driven technique, deep learning relies heavily on labeled ground-truth data to train the model and help it acquire sufficient domain knowledge. Conventional machine learning approaches, such as support vector machines or decision trees, usually require expert-crafted features as input. In contrast, deep learning models autonomously extract these features from raw data, thereby reducing the need for manual feature engineering. Nevertheless, this capability relies on a significant volume

of labeled data to guide the learning process. The more complex the task, the more labeled examples the model typically needs to effectively learn useful patterns.

     In sea ice research, acquiring a large amount of labeled data is often challenging due to several factors, such as logistical issues, the dynamic nature of sea ice, and the high cost of data collection. Some researchers have attempted to tackle these challenges by utilizing data from numerical simulation systems (Liu et al., 2023). These systems leverage mathematical modeling

to predict sea ice behavior and attributes, offering an abundance of data without the need for physical data collection. However, these systems are not without their own issues. Primarily, the simulations carry a degree of uncertainty. This uncertainty stems from the inherent limitations of modeling complex natural processes, inaccuracies in the input data, and the assumptions made in the models. In addition, the simulation systems also require human effort, such as data pre-processing, calibration of the simulation models, and validation of the simulation results against actual observations. This process can be labor-intensive and

demands a high level of expertise.

     In response to this challenge, iterative learning, also known as incremental learning, is a strategy that has been employed in recent literature (Hoffman et al., 2021; Sudakow et al., 2022). This approach starts with the labeling of a small data subset, providing the groundwork for the initial training of the model. Subsequent to the training phase, the model is tasked with



generating label predictions for the remaining unlabeled dataset. Human experts then review these predictions, rectifying
evident errors and annotating notable omissions. The refined predictions are subsequently utilized to train another model
iteration. This collaborative interplay between the model and human experts helps ensure that the labeling process retains a
high level of accuracy while also benefiting from the efficiency of the AI system. Sometimes, some datasets serve as ground
truth for the others. Yin et al. (2021), for instance, derived higher-resolution MODIS data using Landsat-8 as ground truth.
MODIS, while offering high temporal resolution data, is at a coarser spatial resolution (250m to 1km) in comparison to the
finer spatial resolution (30m) offered by Landsat-8, which has a lower temporal resolution. This is achievable due to the
overlapping wavelengths of the two satellite systems, making inter-comparison possible.

Another popular approach is semi-supervised learning, leveraging both labeled and unlabeled data during the training pro-
cess. This method not only utilizes the known labels to guide the learning but also exploits the underlying structure in the
unlabeled data to improve generalization. For instance, similar data points are expected to have similar labels. Jiang et al.
(2022a) used semi-supervised learning for classifying sea ice types. The method began with the Iterative Region Growing
using Semantics (IRGS) process to generate super-pixels. A GCN was then constructed to learn the features of each node
representing these super-pixels, forming the basis for label assignment. Finally, a softmax assigns labels to the nodes in the
graph.

## 3.4 Advanced models and architectures

Deep learning has increasingly become an indispensable tool for researchers in sea ice studies over the past few years, har-
nessing the ability of these models to recognize intricate patterns in large, complex datasets. While conventional deep learning
models have achieved significant success in tasks such as sea ice classification and tracking, they do have their limitations.
Conventional CNN models, for example, often fail to fully utilize contextual information and can struggle with scale variations
in image data. This can lead to less-than-optimal performance when classifying ice types that show similar spectral charac-
teristics but different contextual information. LSTMs, while excellent at capturing long-term dependencies, in practice, they
can still struggle with very long sequences due to vanishing gradients. Moreover, their sequential nature makes it difficult to
parallelize operations during training. In view of these limitations, sea ice researchers are progressively turning towards more
advanced models and architectures.

Embracing the potential of more advanced models, sea ice researchers are now exploring the realm of attention mechanisms
and Transformer architectures. Attention mechanisms allow models to focus on specific parts of the input while processing
data, thereby enhancing the model's ability to use contextual information. This can be particularly useful in sea ice studies,
where distinguishing between similar-looking ice types often hinges on understanding the broader context (Liang et al., 2023;
Ren et al., 2022; Wei et al., 2022). On the other hand, Transformer models, initially designed for natural language processing
tasks, have proven to be remarkably effective in image processing tasks as well, including image segmentation. The self-
attention mechanism within Transformers enables the model to account for all pixels in the input image, thereby addressing
scale variations and enabling a more global understanding of the image. This capability significantly enhances the accuracy of



tasks like sea ice type classification (Sudakow et al., 2022), where classifying each pixel correctly is crucial. Unlike LSTMs, Transformers can process inputs in parallel, improving computational efficiency and facilitating large-scale applications.

Continuing the trajectory of innovation, newer deep learning models such as NFNets (Brock et al., 2021), PredRNN++ (Wang et al., 2018), and MobileNetV3 (Howard et al., 2019) are being incorporated into the toolkit of sea ice researchers, each offering distinct advantages over conventional models like VGGNets and LSTMs. NFNets, a family of models designed to provide high performance without the need for batch normalization, have shown promise in handling large-scale datasets. In the context of sea ice research, the improved performance and scalability of NFNets lead to better accuracy in tasks like sea ice classification (Lyu et al., 2022b), particularly when dealing with high-resolution satellite imagery. The elimination of batch normalization also reduces the computational resources needed during the training process, making the model more efficient to use. PredRNN++, an advanced recurrent model, overcomes the vanishing gradient problem in traditional LSTMs and can handle longer sequences, making it suitable for studying long-term sea ice dynamics. The enhanced ability of PredRNN++ to handle long-term spatio-temporal dependencies and improved performance in prediction tasks lead to more accurate forecasts of sea ice extent and thickness (Liu et al., 2023). MobileNetV3, a highly efficient model architecture optimized for mobile devices, offers the benefit of lightweight structure and lower computational requirements. When combined with a multi-scale network such as FPN (Feature Pyramid Network), the benefits of MobileNetV3 can be further enhanced. FPN can help the model capture and integrate features at various scales, which is often critical for tasks such as sea ice classification, where the ice can display variations in scale (Zhang et al., 2022b).

## 4    Future research directions

AI is a rapidly evolving field, with continuous advances in new methods and techniques. While we are witnessing an increasing number of transdisciplinary research projects applying AI to Arctic sea ice remote sensing, we also see opportunities to further integrate and adapt cutting-edge AI into Arctic sea ice research. Below, we present a few research directions from which the sea ice research community could further benefit, leveraging the most recent and exciting developments in AI.

### 4.1    Enhanced multi-modal deep learning capabilities

Multi-modal deep learning is an advanced machine learning approach that harnesses information from multiple data types (or "modalities") such as text, images, audio, and video to make more accurate and comprehensive predictions. It involves training deep neural networks on data that includes multiple types of information and creating a unified representation that captures the joint information present in these diverse data. This approach is particularly useful in scenarios where individual modalities might not provide a complete picture, and the integration of different types of data can lead to more robust and insightful models. For example, an image of a scene could be combined with textual descriptions to provide a more comprehensive understanding.

Recent advancements in the field of multi-modal deep learning have further broadened its applicability and effectiveness. For instance, researchers have been developing innovative algorithms that can not only process and integrate different data



types but also account for the temporal dynamics and dependencies between these modalities, such as the synchronization of audio and video data in multimedia content (Korbar et al., 2018; Park et al., 2022; Yang et al., 2017). Moreover, the advent of transformer-based models, such as ViLBERT (Vision-and-Language BERT; Lu et al., 2019), LXMERT (Learning Cross-Modality Encoder Representations from Transformers; Tan and Bansal, 2019), and MMF (MultiModal Framework; Singh et al., 2020), has significantly improved the performance of multi-modal tasks by better understanding and representing the complex interactions between different data modalities. Despite these advances, there remain substantial challenges, including the need for larger, more diverse datasets and methods to better interpret and explain the decisions of these complex models. In parallel with these developments, the sea ice research field has begun to incorporate multi-modal deep learning methodologies into its predictive models. An array of multi-source data, including sea ice data, meteorological data, and satellite-derived datasets, is already being used. Typically, researchers have approached data fusion by aligning them through input channels or processing the data streams separately and then combining them through concatenation (Andersson et al., 2021). This approach, while effective to an extent, still needs further refinement and optimization. The current progress of multimodal deep learning has opened a plethora of research opportunities for sea ice studies to understand and interpret the nuanced interconnections among multi-source data.

Looking towards the future of sea ice research, the potential of multi-modal deep learning to enhance our comprehension and prediction of sea ice transformations is unquestionably promising. A crucial aspect to consider is the necessity for ample, diverse, and high-quality datasets to fuel the evolution of multi-modal deep learning within this research area. When it comes to handling data, the opportunity to align various data modalities is immensely valuable. Given the ever-changing nature of sea ice time-sequence data, temporal alignment has become the key. By combining datasets such as sea ice concentration, satellite imagery, and climate patterns that coincide with spontaneous temporal events, we may unravel intricate relationships that are hidden when examining individual datasets. Moving forward, fusion strategies for multi-modal input that extend beyond basic concatenation or stacking need to be explored (Wang and Li, 2021a). For instance, hierarchical fusion that consolidates information across different levels of a deep learning model could enhance our grasp of sea ice dynamics (Wang et al., 2022b). Likewise, implementing cross-modal attention mechanisms (Ye et al., 2019) could enable the model to adjust its focus across various modalities selectively, emphasizing salient features based on the context. In the realm of multi-modal representation learning (Guo et al., 2019), there exists the possibility to devise innovative techniques to conjointly represent different modalities. Approaches like canonical correlation analysis (CCA) or its deep learning equivalent (DCCA) could optimize the interplay between various data modalities, thereby augmenting these models' predictive power. In addition, the emergence of transformer-based models in the multi-modal learning landscape opens a path for deeper investigation. With the ability to harness the self-attention mechanism of transformers, these models could better discern inter-modal and intra-modal dependencies, resulting in more accurate forecasts of sea ice conditions. While these advancements pose significant challenges, they hold the potential to fundamentally transform our understanding of sea ice dynamics.





## 4.2 Better ability to quantify uncertainty

Quantifying uncertainty in the context of deep learning refers to the process of estimating the confidence of a model's predictions. This is important because first, it allows the users of the model to understand the level of risk associated with each prediction, enabling them to make informed decisions. Second, it can lead to improved model performance by highlighting
areas of the input space where the model is uncertain and may need additional training data. Third, it contributes to the development of more robust and trustworthy AI systems, as models that can quantify their uncertainty provide more transparent and interpretable outputs. This uncertainty can arise from several sources including the inherent noise in the data, incomplete or biased data, and model limitations such as overfitting or underfitting.

In the realm of deep learning, significant strides have been made in the field of uncertainty quantification. A key develop-
ment has been the incorporation of Bayesian methods into neural networks, leading to what is commonly known as Bayesian Neural Networks (BNNs; Blundell et al., 2015; Gal and Ghahramani, 2016; Goan and Fookes, 2020). These networks offer a principled way of encoding uncertainty by treating model parameters as random variables, allowing for the computation of a posterior distribution over these parameters. Ensemble methods have also proven effective in quantifying uncertainty. By training multiple models and comparing their outputs, these methods can provide an estimate of the variation in predictions,
offering a measure of uncertainty. This has been seen in Deep Ensembles (Lakshminarayanan et al., 2017), a technique that involves training multiple deep networks independently and then combining their predictions. Furthermore, advancements have been made in using out-of-distribution detection methods to quantify uncertainty (Delaney et al., 2021; Everett et al., 2022). These techniques aim to identify when a model is making predictions on data that significantly differs from its training data, a situation where the model's predictions are inherently uncertain. This is also a test of the model's generalizability (Good-
child and Li, 2021). Finally, research on uncertainty quantification has influenced the development of new loss functions, like the Variational Autoencoder's (VAE) evidence lower bound (ELBO; Kingma et al., 2019; Kingma and Welling, 2013) or the Wasserstein loss for Generative Adversarial Networks (GANs; Arjovsky et al., 2017; Frogner et al., 2015), which balance the reconstruction accuracy with the uncertainty in the model's predictions. While these methods have marked considerable progress, uncertainty quantification in deep learning remains a challenging and active area of research.

Uncertainty quantification in sea ice research utilizing deep learning is a burgeoning field with vast research potential, as evidenced by a limited but innovative body of work. One pioneering approach involves integrating uncertainty estimation directly from the dataset into the model's training process. This is achieved by scaling the loss value based on the level of uncertainty, thereby allowing the model to better account for variability in the data and potentially enhancing its overall performance (Dominicus et al., 2021). Another methodology is the implementation of an aleatoric uncertainty-embedded transfer
learning approach for SAR sea ice classification. This technique involves placing a multivariate Gaussian distribution onto the embedding of input data in the latent space to model aleatoric uncertainty, a form of uncertainty that arises from inherent noise in the data (Liu et al., 2022b). Despite these advances, numerous challenges and opportunities remain. Uncertainty in sea ice research can arise from multiple sources, including but not limited to, the equations used for calculating SIC, the variability in numerical models, and the precision of the instruments employed. Furthermore, inherent limitations in deep learning





models, such as overfitting or underfitting, can also contribute to uncertainty. The need to account for these diverse sources of uncertainty presents numerous research opportunities.

Moving forward, future research directions in sea ice research should emphasize the development and application of ensemble deep learning methods for uncertainty quantification. Ensemble methods, such as bootstrapped ensembles or Bayesian neural networks, offer a robust avenue for quantifying predictive uncertainty. They can leverage the power of multiple inde-
pendently trained models to capture a broad range of uncertainties, including epistemic uncertainty, which arises from lack of knowledge or data. This could provide a more comprehensive uncertainty estimation, enhancing the credibility and reliability of sea ice predictions. In addition, the application of deep learning for uncertainty quantification in sea ice research could benefit from the integration of physical laws and principles. Known as physics-informed machine learning, this approach can help constrain model predictions and provide insights into the sources of uncertainty. For instance, physical laws, such as the
conservation of energy, could be incorporated into the loss function of the deep learning models, thereby reducing model bias and uncertainty. Another promising direction is the use of explainable artificial intelligence (XAI; Hsu and Li, 2023) techniques in deep learning-based sea ice research. The interpretability of deep learning models has been a long-standing challenge, and the lack of transparency can exacerbate uncertainty issues. Applying XAI techniques can help uncover how these models make predictions, potentially leading to a better understanding of the uncertainty and its sources. Lastly, integrating deep learning
with traditional statistical methods for uncertainty quantification, such as Monte Carlo methods or Bayesian inference, can provide a more comprehensive approach to handling uncertainty. This kind of hybrid approach could leverage the strengths of both deep learning and traditional statistical methods, providing a powerful tool for uncertainty quantification in sea ice research.

## 4.3 Better leverage of AI foundation models

AI foundation models are large-scale machine learning models that are pre-trained on a vast amount of data and can be fine-tuned for various downstream tasks. The term, often associated with language models like GPT-4 (OpenAI, 2023), image segmentation models like Segment Anything Model (SAM; Kirillov et al., 2023), describes models that serve as a general basis for many different AI applications, offering a kind of "foundational" intelligence that can be built upon. The importance of AI foundation models lies in their generalizability and efficiency. By pre-training a model on a large dataset, it learns a broad
base of knowledge that can be applied to many different tasks with very modest additional training efforts, a.k.a., fine-tuning. This saves time and resources compared to training a model from scratch for each new task. Additionally, these models often exhibit strong performance and domain adaptability across a wide range of tasks, in some cases even reaching or exceeding human-level performance.

Building upon this, AI foundation models can be categorized into several types, each with its unique functionality. Language
models like GPT-4 (OpenAI, 2023) and LLaMA (Touvron et al., 2023), trained on extensive text data, excel at generating human-like text and can be used for a wide array of tasks such as translation, summarization, and question-answering. Image models like SAM (Kirillov et al., 2023) and Florence (Yuan et al., 2021), trained on image data, are adept at recognizing and segmenting objects in images, classifying images, and even generating images. There are also multimodal models that can



handle tasks necessitating the understanding of multiple modes of data (Alayrac et al., 2022; Radford et al., 2021). Foundation
models' functionality is rooted in a two-step process—pre-training on extensive data in an unsupervised or self-supervised
manner to learn useful data representations, followed by fine-tuning on task-specific data to optimize performance.

In the realm of sea ice research, the potential for deploying AI foundation models remains largely untapped, presenting a
promising area for exploration and development. While no definitive foundation model for sea ice area has been established
yet, the groundwork has been laid with the advent of several models focused on remote sensing imagery. These include the
work by Cha et al. (2023), which demonstrated the scalability of foundation models and was developed for detecting and
segmenting various man-made features in satellite or aerial images, and RingMo (Sun et al., 2023), which proposed a large
dataset of two million remote sensing images and trained a model to navigate dense, complex remote sensing scenes with small
objects. Another notable model is proposed Wang et al. (2022a), who introduced a new technique called rotated varied-size
window attention (RVSA) to manage the large image sizes and objects of various orientations commonly found in remote
sensing imagery. The existence of these models indicates a favorable momentum towards the creation of a sea ice foundation
model. Furthermore, the interconnected nature of sea ice tasks—ranging from ice type classification to SIC classification/fore-
casting—suggests that a foundation model could serve as a highly effective tool, able to generalize across different tasks and
offer significant acceleration in sea ice research and new discoveries.

Considering the significant progress already made, there are several plausible directions to consider for establishing a foun-
dation model in sea ice research. First, leveraging the wealth of available data—encompassing satellite imagery, SIC data, and
meteorological data—is key. This data could be used to pre-train a versatile foundation model that can handle multiple sea
ice-related tasks. Furthermore, the previous foundation model works (Cha et al., 2023; Sun et al., 2023; Wang et al., 2022a)
provide valuable insights for this endeavor. For instance, incorporating techniques like the RVSA could help the model effec-
tively manage large image sizes and objects of various orientations, common in remote sensing imagery. In addition, adopting
the scaling strategy demonstrated by Cha et al. (2023) could further enhance the robustness and performance of a sea ice
foundation model. Additionally, a focus on multimodal learning could be beneficial, given the diverse nature of data involved
in sea ice research. A model capable of processing and integrating information from different types of data—such as visual
data from images and numeric data from meteorological measurements—could provide a more holistic understanding of sea
ice conditions and dynamics. As the very first step, sea ice researchers could try to adapt existing AI foundation models, such
as SAM and the new IBM's Prithvi (Jakubik et al., 2023), and fine tune them for problem solving related to sea ice. Thinking
more broadly, there is the prospect of building a foundation model that supports general Arctic research, trained with massive
amounts of Arctic data with multi-modality. This effort will benefit the Arctic and polar communities as a whole.

## 4.4 Deeper integration with physics-based models

Physics-based deep learning models are a fusion of traditional physics-based modeling and modern deep learning techniques.
In this approach, deep learning models are designed to adhere to known laws of physics or to integrate these principles directly
into their architecture or learning process. The importance of physics-based deep learning models lies in their ability to leverage
existing knowledge about the world to improve predictions. For instance, in climate science, physics-based models can use the



laws of thermodynamics to better simulate climate patterns. By incorporating known physical laws, these models can make more accurate predictions, generalize better to unknown situations, and sometimes even learn more efficiently, as they're not

starting from scratch but building on a foundation of established understanding. They can also provide more interpretable predictions because the outputs are tied to physically meaningful mechanisms or principles. These models can make better predictions in situations with sparse or noisy data, as they can leverage prior physical knowledge to "fill in the gaps."

The field of physics-based deep learning models is a rapidly evolving area with numerous contributions (Thuerey et al., 2021). One of the first works in this space was the use of CNNs in fluid dynamics (Guo et al., 2016; Morimoto et al., 2021),

where the networks were taught to predict the time evolution of fluid flows with high accuracy, respecting the physical laws of fluid dynamics. In another seminal work, researchers integrated the Hamiltonian function, a concept from classical physics, directly into the architecture of a neural network, giving rise to Hamiltonian Neural Networks (HNNs; Greydanus et al., 2019). This enabled the model to learn and predict complex dynamic systems accurately while inherently conserving energy, a fundamental principle in physics. The concept of physics-informed neural networks (PINNs) is another significant contribution to

this field (Raissi et al., 2019). PINNs incorporate differential equations, representing physical laws, directly into the loss function of a deep learning model. This allows the model to learn solutions that inherently follow the given physical laws. PINNs have been applied to a wide range of problems, from solving partial differential equations to modeling wave propagation and heat conduction. More recently, the notion of "deep equilibrium models" (DEQs) has gained traction (Bai et al., 2019). These models are designed to find the equilibrium state of a physical system directly, rather than simulating its temporal evolution.

DEQs leverage the power of deep learning to solve complex equilibrium equations that would be difficult or impossible to solve analytically. Moreover, physics-based deep learning models have also been used to tackle the issue of data scarcity and noise (Bai et al., 2022). The integration of physical laws into these models allows them to make robust predictions even with sparse or noisy data.

Despite the promising advancements in the field of physics-based deep learning models, there has been a noticeable lack

of such research specifically in the area of sea ice studies. In one of the few instances of such work, deep learning has been employed to improve sea ice classification performance through the creation of physics-aware training data (Wang et al., 2022c). The majority of deep learning-based sea ice research, however, relies heavily on data derived from physics-based simulation models such as ERA5. These datasets provide essential environmental parameters that are integral to understanding and predicting sea ice dynamics. It is here that a significant opportunity lies - to marry the strengths of physics-based models

and deep learning. Physics-based models excel in their ability to simulate the complex interactions and processes in the sea ice environment, albeit often with some uncertainty due to approximations and assumptions in the model. On the other hand, deep learning models, with their ability to learn complex patterns from large volumes of data, can potentially enhance the predictive capabilities of these physics-based models. Furthermore, the incorporation of physical laws, as seen in the concepts of PINNs and HNNs, could provide a robust framework for dealing with the challenges inherent in sea ice prediction, such

as data scarcity, noise, and the need to maintain conservation laws. Thus, the underutilized potential of physics-based deep learning models in sea ice research presents a promising avenue for future exploration and innovation.





Given the groundwork already laid by the application of physics-based deep learning models in earth science, sea ice researchers stand at the threshold of exciting new directions for future research. First and foremost, the application of PINNs in sea ice studies could be explored. PINNs have shown promise in various areas of earth science, and their capacity to integrate physical laws directly into the learning process could provide a powerful tool for modelling and predicting sea ice dynamics. Similarly, Hamiltonian Neural Networks (HNNs; Greydanus et al., 2019), with their focus on energy conservation, could offer unique insights into the energy balance processes that govern sea ice formation and melting. Furthermore, researchers could look to Deep Equilibrium Models (Bai et al., 2019) as a potential framework for investigating the equilibrium states of sea ice under various atmospheric and oceanic conditions. Given that many earth science applications involve finding equilibrium states, the transfer of this concept to sea ice dynamics could be highly beneficial. The high degree of similarity and overlap between earth science and sea ice research, coupled with the potential of physics-based deep learning, presents an exciting frontier for future research and innovation in sea ice studies.

## 5 Conclusions

This review paper provides a comprehensive overview of the transformative role that artificial intelligence (AI) has played in the field of Arctic sea ice remote sensing. The application of AI techniques has opened up new horizons for researchers, enabling them to extract invaluable insights from the vast and dynamic Arctic region. Throughout this paper, we have highlighted the key contributions of AI, especially deep learning, in various aspects of sea ice remote sensing, from sea ice lead detection, sea ice thickness estimation, to sea ice concentration forecasting, and sea ice motion prediction. Compared to traditional machine learning, such as random forest and support vector machines, deep learning models possess outstanding capability in enabling large-scale automated study and better capturing spatial and temporal dependencies within the datasets. These novel AI applications have demonstrated superior performance in improving the accuracy and efficiency of sea ice classification, detection, and forecasting.

In addition to using off-the-shelf AI and deep learning models, we have also witnessed an increase in the development of customized learning strategies, such as self-attention and new loss functions, to tailor AI models to advance sea ice research. There is also a rise in adopting cutting-edge AI models, such as transformers, to solve sea ice-related problems at a pan-Arctic scale. Built upon the momentum of adopting AI in Arctic sea ice research, we suggest future research directions where the Arctic sea ice research community could further benefit from the latest advancements of AI. This includes key research questions such as uncertainty quantification and a deeper integration of physics-informed sea ice modeling and deep learning, which are currently under investigated.

Looking ahead, the future of AI for Arctic sea ice remote sensing holds great promise. As technology advances and more data become available, AI models will continue to evolve, enabling more accurate predictions, real-time monitoring, and an enhanced understanding of the Arctic ecosystem. This, in turn, will contribute to our global efforts to address climate change, ensure safer navigation in polar regions, and support sustainable resource extraction. In conclusion, the fusion of AI and Arctic sea ice remote sensing is a testament to the incredible progress we have made in understanding one of the Earth's



760 most vulnerable and critical environments. It is our hope that this review paper serves as both a guide and an inspiration for researchers, engineers, and policymakers to further explore the limitless potential of AI in the service of the Arctic and our planet. Together, we can harness the power of AI to protect and preserve this unique and vital region for generations to come.

*Author contributions.* WL conceived the idea; WL, CH, MT discussed the paper; WL and CH wrote the manuscript draft; MT reviewed and edited the manuscript.

765 *Competing interests.* The authors declare that they have no conflict of interest.

*Acknowledgements.* This work is supported in part by the National Science Foundation under award 2230034.



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
