# Peer review of "Advancing Arctic sea ice remote sensing with AI and deep learning: now and future"

_EGUsphere, 2023_

## Referee Comment (RC1)

**Review of egusphere-2023-2831**

Dear authors of the manuscript egusphere-2023-2831 titled "Advancing Arctic sea ice remote sensing with AI and deep learning: now and future". Use of Artificial Intelligence (AI) and Machine Learning (ML), especially ML based on deep learning (DL), in sea ice remote sensing and number of related publications have rapidly increased during the recent years and the results have been very convincing. From this point of view reviews of the field are welcome and very useful for the sea ice remote sensing research and operational sea ice monitoring. This manuscript includes much valuable information on different ML/DL approaches for remote sensing of sea ice. However, the manuscript will still need to be revised, the scope needs to be refined and some concepts to be clarified. Therefore, I propose a major revision before publishing the manuscript. In the following are my comments and especially, be careful with the terminology and concepts used in the manuscript and the focus of the manuscript.

**General comments:**

Title and Scope: Now the title includes Arctic sea ice remote sensing. Why only Arctic is included? Consider changing the geographical area to include all sea ice (in practice Arctic and Antarctic), consider dropping Arctic from the title. The title also includes "AI and deep learning". This indicates that both AI in general and deep learning (DL) will be discussed in the manuscript. Regarding to concept AI: ML is a subset of AI and DL is a subset of ML. Based on the title this manuscript should then include all AI/ML-based Arctic sea ice remote sensing, also earlier methods before wide application of DL (such "earlier" methods are for example Support Vector Machines, Random Forests etc.) but they have not been included, even though good results have been achieved also using these methods. I suggest to restrict the methodological scope already in the title, e.g. by dropping AI and also emphasizing the methodological scope in the introduction section, including a good reasoning for the restricted scope, otherwise the manuscript should be very long to cover what the tile suggests. There is also the word "future" in the title. It is very difficult to predict the future, maybe "future" could also be dropped or modified somehow, e.g. "future prospects" or something like that referring to possible/potential future developments instead of trying to provide too deterministic future predictions.

Abstract: It is mentioned that the manuscript will provide a comprehensive review. Still, there are many missing things (e.g. ice deformation, ice age) to complement the topics mentioned in the abstract and elsewhere in the manuscript.

Structure: In general structure looks reasonable. Section 2 needs some introductory text before starting subsection 2.1. Some subsection titles need to be changed/updated.

Sea ice details: for example, detection of pressure ridges and rubble fields and ice floe distribution are not at all or very shortly discussed, more related information should be included.

Ice motion: from ice drift many related properties can be derived, such as divergence/opening (development of leads, cracks) convergence and shear (ice deformation, ridging). Based on momentun of the ice motion also short-tern prediction is possible based on estimated ice drift information. These things should at least be mentioned.

XAI is mentioned in the uncertainty-related section. However, potential of XAI for interpreting the AI models and for improving the understanding and development of physical model based algorithms based on XAI analysis has not much been discussed. This is an interesting and emerging topic and should be discussed more. Any new publications on XAI applied sea ice remote sensing

data? I included information on a couple of general level publications on XAI at the end of this review. Consider including these references when presenting XAI in the manuscript.

Qualitative and quantitative comparison to earlier quite recent reviews (e.g. Zakhvatkina et al., 2019) should be included. This e.g. reveals the recent rapid increase of deep learning approaches in the field. It would also be useful to include some numbers comparing the performance of the state-of-the-art methods using DL and the earlier state-of-the-art algorithms for sea ice parameters with reference data available (e.g. SIC and SIT estimate accuracy etc.).

In earlier research and publications sea ice imagery, especially SAR imagery, segmentation without classification was one topic, for example using Markov Random Fields.  The current DL algorithms typically perform a semantic segmentation, i.e. combining classification with the segmentation. SAR segmentation can be used for example as an input to ice charting instead of manually drawn polygons. The attributes describing the ice within each segment can then be assigned to the segments after the segmentation either manually or automatically. Are there any novel DL publications on this topic? This topic should be mentioned in any case.

Data fusion from multiple EO sources, such as altimeters and SAR, SAR and ice models etc., should be discussed more. This topic suits well into the subsection on multimodal learning.

Combination of continuous development data processing methods and  EO instruments and their increased amount should be discussed more. Possible future development is e.g. fusion of nearly simultaneous SAR imagery of multiple frequencies (e.g. X, C and L bands).

For example, detection of sea ice details, such as pressure ridges or rubble fields and ice floe size distribution are not discussed much. These are important topics and become easier to locate from EO data as the resolution and data quality of instruments increases. There should be more discussion related to this topic.

ML/DL hyperparameter optimization for sea ice: this topic is not discussed (enough), even though it is an important topic for both algorithm accuracy and computational performance. Are there any recent related publications?

Reference datasets: There exist some publicly available large reference datasets with EO data and the corresponding labeled ground-truth data. At least, the Danish Meteorological Institute (DMI) and Danish Technical University (DTU) EO4Arctic dataset (Buus-Hinkler et al., 2022) is one, there may be more(?). The existence of such datasets is important for the algorithm development and comparison. This should be emphasized and references to available reference datasets given.

Update the paper and references with the most recent publications published after submission of the first version. Number of publications on Sea ice remote sensing using ML and DL is increasing at a furious pace.

**More detailed comments:**

I won't go into every detail at this phase of the review process because it does not make much sense. I'll provide more detailed comments after the possible major review. Here are some detail comments:

Abstract:
P1 L5: "lead detection" → "lead and deformation detection". Here I by deformation refer to pressure ridges and rubble fields.

P1 L5: "concentration" → "concentration estimation"

Introduction:
P1 L19: Open IPCC

P2 L34: A good general reference to MLP with error backpropagation is e.g. Rumelhart et al., 1986. Include this reference.

P2 L55: "predict" typically refer to predicting future, preferably use "estimate" or maybe "assign" here .

P3 Fig. 1. I suggest to replace "sea ice extent forecasting" by "sea ice extent estimation and forecasting", in the same context could also be "sea ice detection" included as sea ice extent is closely related to sea ice detection. "sea ice lead detection" could be replaced by "sea ice detail detection" to include e.g. pressure ridges and other forms of ice deformation also. Consider also including "sea ice segmentation" in the figure.

P3L67: "...enhance their detection and movement".- This is a strange sentence. Do the leads really move, Rather the ice is moving around them,. Preferably say something like "enhance the detection of ice ridges and their changes" or something that makes (even) better sense.

AI and deep learning applications to Arctic Sea ice research:
Consider changing the title e.g. to "DL applied to sea ice remote sensing" or something similar. Include some introductory text before the subsection 2.1.

P8L190: I suggest to replace the title by "SIC estimation", even though classification to quantized SIC categories is typically used in this context, the fundamental purpose is to estimate the physical parameter SIC as accurately as possible (taking into account the restrictions of the reference data).

P10L246: "Forecasting the change in sea ice extent". If there is a section on forecasting sea ice extent, then there should also be text on the time series used for training these forecasting algorithms. As this is a remote sensing manuscript, the time series should be based on remote sensing data, preferably based even on DL methods. Consider including a section on time series of estimated sea ice properties (SIT, SIC). The (joint) section name could then be "forecasting of changes in sea ice based on remote sensing (history data)". Anyway, the remote sensing part of prediction restricts to the training and evaluation data, in prediction we have to rely on models (DL based models in this case).

P12-14, Sea ice motion estimation. Include here also information of the quantities derived from the estimated ice motion (convergence, divergence, shear, causing opening and deformation). Say something on possibilities of multitemporal image analysis in sea ice classification in general. Possible accuracy comparison between "traditional" methods (maximum cross-correlation, feature matching, optical flow) and novel DL methods, based on earlier published results, would be useful.

Deep learning strategies tailored for Arctic Sea ice research:

I propose to change the title to "Customized DL for sea ice characterization (from remote sensing data)" or something similar. I would drop "Arctic" to make the scope more general and also "research" because these methods can and will be used also for operational purposes (e.g. to aid safe navigation and offshore activities), not only for research.

Subsection 3.2. Include the reference(s) to Kucik & Stokholm, 2023 in this section.

Subsection 3.3.. Include the reference Karvonen 2021 demonstrating a way to generate SIC training samples by combining open water and sea ice blocks.

Section 3.4. Is "advanced" a suitable word here? I suggest to use e.g. "novel", "recent" or "evolving" instead in the subsection title and in the text. In my opinion also the methods presented earlier are quite advanced. If the purpose is to indicate that the models presented here are more structurally and computationally complex, then use a better suitable word.

Future research directions:

I propose to use "Potential future research directions" here because predicting the future is not so straightforward. In the future there will be more instruments, such as SAR instruments operating at multiple frequency bands (X, C. L), the joint use of multi-frequency SAR and also data fusion with other available data (e.g. altimeters and ice models) will be rather obvious future development steps. Data fusion of radar altimeter and SAR with a more "traditional" method for SIT estimation has been studied e.g. in Karvonen et al., 2022. There may also be some related publications applying DL also? Combining SAR and microwave radiometer data for SIC estimation has already been studied in many publications. Also fusion of sea ice models and remote sensing data should be mentioned.

Application of XAI should be mentioned here, both in sections 4.3. and 4.4. XAI can be used to better analyze what the AI/DL is learning and analyze the algorithm. This can then also lead to improved explanations of physical models (section 4.4). Optimization, optimization methods and strategies of the DL hyperparameters is also a developing field that should be discussed in the manuscript.

Conclusions: This section is quite short. Consider including some conclusions on the presented/referred methods that perform well for certain sea ice observation tasks and how they differ (in terms of performance and computational complexity/requirements) from the earlier methods for measuring sea ice properties based on EO data. You may also indicate conclusions on the most promising approaches for certain purposes and their possible/expected evolution in the near future.

**Some references to be considered to be included in the manuscript:**

N. Asadi, K. A. Scott, A. S. Komarov, M. Buehner and D. A. Clausi, "Evaluation of a Neural Network With Uncertainty for Detection of Ice and Water in SAR Imagery," in *IEEE Transactions on Geoscience and Remote Sensing*, vol. 59, no. 1, pp. 247-259, Jan. 2021, doi: 10.1109/TGRS.2020.2992454 (including uncertainty)

Karvonen, J., Baltic Sea Ice Concentration Estimation From C-Band Dual-Polarized SAR Imagery by Image Segmentation and Convolutional Neural Networks, IEEE Transactions on Geoscience and Remote Sensing (Early Access), DOI: 10.1109/TGRS.2021.3097885, 2021 (synthesized SIC training data)

Park, J.-W., Korosov, A. A., Babiker, M., Won, J.-S., Hansen, M. W., and Kim, H.-C.: Classification of sea ice types in Sentinel-1 synthetic aperture radar images, The Cryosphere, 14, 2629–2645, https://doi.org/10.5194/tc-14-2629-2020, 2020 (Random Forest classification)

Zakhvatkina, N.; Smirnov, V.; Bychkova, I. Satellite SAR Data-based Sea Ice Classification: An Overview. *Geosciences* **2019**, *9*, 152. https://doi.org/10.3390/geosciences9040152 (an earlier review of SI classification, also including some NN related references)

Karvonen, J., Rinne, E., Sallila, H., Uotila, P., and Mäkynen, M.: Kara and Barents sea ice thickness estimation based on CryoSat-2 radar altimeter and Sentinel-1 dual-polarized synthetic aperture radar, The Cryosphere, 16, 1821–1844, https://doi.org/10.5194/tc-16-1821-2022, 2022. (data fusion)

Kucik, A., Stokholm, A. AI4SeaIce: selecting loss functions for automated SAR sea ice concentration charting. *Sci Rep* **13**, 5962 (2023). https://doi.org/10.1038/s41598-023-32467-x (loss function selection for SI SAR)

Kucik, A., Stokholm, A. AI4SeaIce: Comparing Loss Foss Representations for SAR Sea Ice Concentration Charting, Proc. ICLR AI for earth and space science workshop 2022.

Zhang T, Yang Y, Shokr M, Mi C, Li X-M, Cheng X, Hui F. Deep Learning Based Sea Ice Classification with Gaofen-3 Fully Polarimetric SAR Data. *Remote Sensing*. 2021; 13(8):1452. https://doi.org/10.3390/rs13081452 (Chinese SAR instruments)

Buus-Hinkler, Jørgen; Wulf, Tore; Stokholm, Andreas Rønne; Korosov, Anton; Saldo, Roberto; Pedersen, Leif Toudal; et al. (2022). AI4Arctic Sea Ice Challenge Dataset. Technical University of Denmark. Collection. https://doi.org/10.11583/DTU.c.6244065.v2 (AI4Arctic dataset)

Rumelhart, David E, Hinton, Geoffrey E and Williams, Ronald J. "Learning representations by back-propagating errors." *nature* 323 , no. 6088 (1986): 533--536. (MLP and error backpropagation)